# Neural DAEs: Constrained neural networks

## Abstract

In this article we investigate the effect of explicitly adding auxiliary trajectory information to neural networks for dynamical systems. We draw inspiration from the field of differential-algebraic equations and differential equations on manifolds and implement similar methods in residual neural networks. We discuss constraints through stabilization as well as projection methods, and show when to use which method based on experiments involving simulations of multi-body pendulums and molecular dynamics scenarios. Several of our methods are easy to implement in existing code and have limited impact on training performance while giving significant boosts in terms of inference.

## 1 Introduction

Many scientific simulations of dynamical systems have natural invariants that can be expressed by constraints. Such constraints represent conservation of some quantities of the system under study. For example, in molecular dynamics, bond lengths between atoms are assumed fixed. Another example is incompressible fluid flow, where the divergence of the velocity field vanishes at any point in space and time. Similarly, in Maxwell's equations, the divergence of the magnetic field vanishes (no magnetic charge). Such additional information about the flow can be crucial if we are to keep the simulations faithful. As a result, a wealth of techniques have been proposed to conduct simulations that obey the constraints at least approximately (Weiglhofer, 1994; Ascher & Petzold, 1998; Allen et al., 2004).

In recent years, machine learning based techniques, and in particular deep neural networks, have been taking a growing role in modelling physical phenomena. In some cases, such techniques are used as inexpensive surrogates of the true physical dynamics and in others they are used to replace it altogether (see e.g. Wang et al. (2018); Degiacomi (2019); Miyanawala & Jaiman (2017)). These techniques use the wealth of data, either observed or numerically generated, in order to "learn" the parameters in a neural network, so that the data are fit to some accuracy. The network is then expected to perform well on new data, outside of the training set and yield simulation results that are accurate and reliable, in many cases, at significantly less computational effort.

From classical simulations, we know that it is often as important to accurately obey the additional constraint information as it is to accurately satisfy the underlying ordinary/partial differential equation (ODE/PDE) system. Nonetheless, no neural network architecture known to us is designed to honour such constraints or invariants. The hope in standard training procedures is that by fitting the data, the network will "learn" the constraints and embed them in the weights implicitly. This, however, has been demonstrated to be insufficient in many cases (Wah & Qian, 2001). As we show in this paper, on some very simple examples, neural networks may be able to approximately learn the dynamics but they can drift off the constraints manifold. This leads to erroneous results that violate simple underlying physical properties. The question then is: *How should additional constraints information be incorporated in a neural network architecture so that this physical information is, at least approximately, honoured?*

The idea of adding constraints information to a network is essentially a continuation of the ongoing process of connecting mathematics with machine learning and explicitly adding known information into a neural network rather than having the network implicitly learn it (Willard et al., 2021). Equivariant networks are one example of this, where the symmetry of a problem is explicitly built into the neural network (Thomas et al., 2018). Previous work on adding constraints to a neural network includes Stewart & Ermon (2017); Xu et al. (2018) which add an auxiliary regularization term to the loss function; Raissi et al. (2019) physics informed networks shape the output of the neural network to fulfill a particular partial differential equation; while Li & Srikumar (2019) incorporates first order logic directly into the neural network.

In this paper we introduce a new paradigm in neural network architectures by developing methodologies that allow us to incorporate such additional information on a dynamical system. Motivated by the field of differential algebraic equations (DAE), we study two different approaches for the solution of the problem. The first is the incorporation of the constraints into the network by Lagrange multipliers, and the second employs so called stabilization techniques that aim to penalize outputs that grossly violate the constraints. Both approaches have similar counterparts in the physical simulation world, and in particular in systems of DAEs (Ascher & Petzold, 1998). Our methodology is designed for residual neural networks, however it can be used and adopted for other architectures as well. We experiment with it on a number of well-known problems, focusing on molecular dynamics (MD) applications (Allen et al., 2004), which enables us to incorporate a variety of constraints often resulting in significant performance improvement.

The rest of this paper is organized as follows: Section 2 describes various ways that constraints can be introduced in neural networks. Section 3 introduces our model problems, and describes relevant constraints for those problems, while Section 4 performs a series of experiments based on those problems using constrained neural networks. The paper is wrapped up in Section 5 with a discussion and conclusions.

## 2 RESIDUAL NETWORKS AND CONSTRAINTS

We consider the problem of statistical learning where we have the data pairs $(\mathbf{x}_i, \mathbf{y}_i), i = 1, \ldots n$, and we assume that $\mathbf{x} \in \mathscr{X}$ and $\mathbf{y} \in \mathscr{Y}$. Our goal is to find a function $f$ depending on parameters $\boldsymbol{\theta}$ that satisfies

$$f(\mathbf{x}, \boldsymbol{\theta}) = \mathbf{y}. \tag{1}$$

We focus our attention on functions $f(\cdot, \cdot)$ that are mapped by neural networks. Such functions typically contain a sequence of multiplications by "learnable" matrices followed by nonlinearities. In this work we particularly focus on the continuous form of residual network architectures that reads

$$
\begin{align}
\mathbf{z}_0 &= \mathbf{K}_o \mathbf{x}, \tag{2a} \\
\dot{\mathbf{z}} &= \sigma(\mathbf{z}, \boldsymbol{\theta}(\tau)), \quad \tau \in [0, 1], \tag{2b} \\
\mathbf{y}_c &= \mathbf{K}_c \mathbf{z}(1). \tag{2c}
\end{align}
$$

The matrix $\mathbf{K}_o$ typically embeds input data $\mathbf{x} \in \mathscr{X}$ in a vector $\mathbf{z}$ in a larger space $\mathscr{Z}$. Next, the residual network uses $m$ layers with learnable parameter $\boldsymbol{\theta}$. Such parameters can be different weights, normalizations and attention parameters. Finally, the larger space is closed by a learnable closing matrix $\mathbf{K}_c$.

For the class of problems we solve here, we assume that there is a given vector function $\mathbf{c}(\cdot)$ such that, $\mathbf{c}(\mathbf{y}_c) = \mathbf{c}(\mathbf{K}_c \mathbf{z}(1)) = 0$. The ODE equation 2b with the initial condition equation 2a represent a trajectory, $\mathbf{z}(\tau)$ in the high dimensional space $\mathscr{Z}$. This trajectory can be projected into the low dimensional, physical space, $\mathscr{Y}$ setting, $\mathbf{y}(\tau) = \mathbf{K}_c \mathbf{z}(\tau)$ which is assumed to be constrained. Therefore, along trajectories $\mathbf{z}$ satisfying equation 2c

$$\mathbf{c}(\mathbf{K}_c \mathbf{z}) = 0. \tag{3}$$

Given the output $\mathbf{y}_c$ one trains the network by first discretizing the ODE (typically by the forward Euler method) and then, minimizing a loss function that measures the difference between $\mathbf{y}_c$ and $\mathbf{y}$. However, even when such parameters are learned, they rarely yield zero loss, especially on the validation set. Therefore, in general, the constraints $\mathbf{c}(\mathbf{K}_c \mathbf{z})$ are not automatically fulfilled and this can yield results that are physically infeasible.

Our goal is to modify the architecture given in equation 2 such that the additional information given by equation 3 is addressed. We next discuss four such approaches that can be used.

### 2.1 AUXILIARY REGULARIZATION

The simplest method for incorporating constraint information is to add an auxiliary regularization term to the loss function. Let $\mathscr{L}$ be the traditional loss. We extend this by adding a regularization term

$$\mathscr{L}_\eta = \mathscr{L} + \frac{\eta}{2} \mathbf{c}(\mathbf{y}_c)^\top \mathbf{c}(\mathbf{y}_c), \tag{4}$$

where the positive parameter $\eta$ determines the strength of the regularization.

While it is possible to use auxiliary regularization successfully as evidenced by Stewart & Ermon (2017); Xu et al. (2018), we have found that auxiliary regularization by itself did not boost a network's learning significantly, as shown in Section C. Furthermore, we have found that the network may still not satisfy the constraints for the validation set, and thus can fail to generalize. Finally, it can be difficult to tune the parameter $\eta$ to have a meaningful balance between the data fit and the constraint.

## 2.2 PROJECTING THE FINAL STATE ONTO THE CONSTRAINT MANIFOLD

Another approach that has been used for problems in image processing where the output is bounded, is to project the final state onto the constraint manifold. To this end, the network is used as is, obtaining a vector $\mathbf{y}_c$. We then look for a perturbation $\delta\mathbf{y}$ such that

$$\mathbf{c}(\mathbf{y}_c + \delta\mathbf{y}) = 0.$$

This problem is not well-posed as there are many such perturbations. Therefore, we look for the perturbation with minimal norm, which results in the iteration

$$\mathbf{y}_{c_{j+1}} = \mathbf{y}_{c_j} - \mathbf{B}\mathbf{J}(\mathbf{y}_{c_j})^\top \mathbf{c}(\mathbf{y}_{c_j}), \quad j = 0, 1, \ldots, M, \tag{5}$$

where $\mathbf{y}_{c_0} = \mathbf{y}_c$ is the output of the unconstrained residual neural network, $\mathbf{J}$ is the Jacobian, that is $\mathbf{J} = \nabla_{\mathbf{y}}\mathbf{c}(\mathbf{y})$, and $\mathbf{B}$ is an approximation to $(\mathbf{J}^\top\mathbf{J})^{-1}$.

Since the network in this method observes the constraint only at the end $\tau = 1$, solving for the projection may require many iterations. Nonetheless, it is important to note that there are no learnable weights in the projection. Thus, by using implicit differentiation it is possible to write an analytic backward function and therefore, there is no need to hold all the states when computing the projection in order to compute the derivative. Using the constraint at the end can be thought of as a "final layer" before the results are sent into the loss function.

Projecting the state at the end has an advantage that it is simple and requires minimal changes to any existing code. However, it may have some serious drawbacks. In particular, the dynamics in equation 2 can lead to states that are very far from satisfying the constraint. In this case, the constraint can be difficult to fulfil and the training can be difficult due to very large changes in the last layer. We therefore next explore two techniques that can be applied in order to follow the constraint throughout the network, at least approximately.

## 2.3 PROJECTING THE STATE ONTO THE CONSTRAINT MANIFOLD THROUGHOUT THE NETWORK

To change the architecture we recall that when constraints are added to an ODE one obtains a DAE. To this end another parameter vector function (a Lagrange multiplier) is added to the system and equation 2b is replaced with

$$\dot{\mathbf{z}} = \sigma(\mathbf{z}, \boldsymbol{\theta}(\tau)) + \mathbf{K}_c^\top \mathbf{J}(\mathbf{K}_c\mathbf{z})^\top \boldsymbol{\lambda}, \quad \tau \in [0, 1], \tag{6}$$

while requiring equation 3 to hold. (See Ascher & Petzold (1998) for derivation.)

Consider first using an explicit stepping method to discretize the system with respect to $\mathbf{z}$, obtaining

$$\mathbf{z}_{k+1}^* = \mathbf{z}_k + h\delta\mathbf{q}(\mathbf{z}_k),$$

where $k$ refers to the $k$-th discrete layer in the neural network, and $\delta\mathbf{q}$ is a single discretization of the ODE with respect to $\mathbf{z}$ around $\mathbf{z}_k$. With forward Euler it has the form

$$\delta\mathbf{q}(\mathbf{z}_k) = \sigma(\mathbf{z}_k, \boldsymbol{\theta}_k).$$

However, since forward Euler may have poor stability when nearly imaginary eigenvalues are present, we use the classical Runge-Kutta 4th order (RK4) discretization which can be written as

$$\delta\mathbf{q}_{RK_4}(\mathbf{z}_k) = \frac{1}{6}\left(\mathbf{k}_1(\mathbf{z}_k, \boldsymbol{\theta}_k) + 2\mathbf{k}_2(\mathbf{z}_k, \boldsymbol{\theta}_k) + 2\mathbf{k}_3(\mathbf{z}_k, \boldsymbol{\theta}_k) + \mathbf{k}_4(\mathbf{z}_k, \boldsymbol{\theta}_k)\right)$$

where each $\mathbf{k}_i$ is the Runge-Kutta stage that requires a single application of a resnet.

Next, we project the update onto the constraint by solving for $\boldsymbol{\lambda} = \boldsymbol{\lambda}_{k+1}$

$$\mathbf{z}_{k+1} = \mathbf{z}_{k+1}^* + h\mathbf{K}_c^\top \mathbf{J}(\mathbf{K}_c\mathbf{z}_{k+1}^*)^\top \boldsymbol{\lambda}. \tag{7}$$

To eliminate $\boldsymbol{\lambda}$ we substitute the right hand side into the constraint, obtaining
$$\mathbf{c}(\mathbf{K}_c \mathbf{z}_{k+1}) = \mathbf{c}\left(\mathbf{K}_c(\mathbf{z}_k^* + h\mathbf{K}_c^\top \mathbf{J}^\top \boldsymbol{\lambda})\right) = 0,$$
with $\mathbf{J} = \mathbf{J}(\mathbf{K}_c \mathbf{z}_{k+1}^*)$. We can now approximately solve this equation for $\delta \mathbf{z} = h\mathbf{K}_c^\top \mathbf{J}^\top \boldsymbol{\lambda}$ in the same way we project the constraint. Linearizing we obtain
$$\mathbf{c}\left(\mathbf{K}_c(\mathbf{z}_{k+1}^* + h\mathbf{K}_c^\top \mathbf{J}^\top \boldsymbol{\lambda})\right) \approx \mathbf{c}(\mathbf{K}_c \mathbf{z}_{k+1}^*) + h\mathbf{J}\mathbf{K}_c\mathbf{K}_c^\top \mathbf{J}^\top \boldsymbol{\lambda} = 0,$$
which gives for the Lagrange multiplier in the $k$th layer the expression
$$\boldsymbol{\lambda} = -h^{-1}(\mathbf{J}\mathbf{K}_c\mathbf{K}_c^\top \mathbf{J}^\top)^{-1}\mathbf{c}(\mathbf{K}_c \mathbf{z}_{k+1}^*). \tag{8}$$
Inserting this expression into equation 7 we finally obtain
$$\mathbf{z}_{k+1} = \mathbf{z}_{k+1}^* - \mathbf{K}_c^\top \mathbf{J}^\top(\mathbf{J}\mathbf{K}_c\mathbf{K}_c^\top \mathbf{J}^\top)^{-1}\mathbf{c}(\mathbf{K}_c \mathbf{z}_{k+1}^*), \tag{9}$$
which performs one iteration of a linearized projection of our state onto the constraint. In practice, we perform several such projections iteratively
$$\mathbf{z}_{k+1}^{j+1} = \mathbf{z}_{k+1}^j - \mathbf{K}_c^\top \mathbf{J}_j^\top(\mathbf{J}_j\mathbf{K}_c\mathbf{K}_c^\top \mathbf{J}_j^\top)^{-1}\mathbf{c}(\mathbf{K}_c \mathbf{z}_{k+1}^j), \quad j = 0, 1, \ldots, M, \tag{10}$$
where $j$ is the projection index, and $\mathbf{z}_{k+1} = \mathbf{z}_{k+1}^M$, and $\mathbf{J}_j = \mathbf{J}(\mathbf{K}_c \mathbf{z}_{k+1}^j)$.

Based on this, the residual network corresponding to equation 2 can now be written as
$$\mathbf{z}_0 \quad = \quad \mathbf{K}_o \mathbf{x}, \tag{11a}$$
$$\delta \mathbf{q}_{RK}(\mathbf{z}_k) \quad = \quad \frac{1}{6}\left(\mathbf{k}_1(\mathbf{z}_k, \boldsymbol{\theta}_k) + 2\mathbf{k}_2(\mathbf{z}_k, \boldsymbol{\theta}_k) + 2\mathbf{k}_3(\mathbf{z}_k, \boldsymbol{\theta}_k) + \mathbf{k}_4(\mathbf{z}_k, \boldsymbol{\theta}_k)\right) \tag{11b}$$
$$\mathbf{z}_{k+1} \quad = \quad \text{proj}_{\mathbf{c}}\left(\mathbf{z}_k + \delta \mathbf{q}_{RK}(\mathbf{z}_k))\right), \quad k = 0, 1, \ldots, n-1, \tag{11c}$$
$$\mathbf{y}_c \quad = \quad \mathbf{K}_c \mathbf{z}_n, \tag{11d}$$
with the projection operator defined by
$$\text{proj}_{\mathbf{c}}(\mathbf{z}) = \mathbf{z} + \{\arg \min_{\delta \mathbf{z}} \frac{1}{2}\|\delta \mathbf{z}\|^2 \quad \text{s.t. } \mathbf{c}(\mathbf{K}_c(\mathbf{z} + \delta \mathbf{z})) = 0\}. \tag{12}$$
Here we have introduced the projection to make the network inherently obey constraints at every layer.

In summary, we have derived a method for approximately projecting the output of each layer onto the constraints. The projections depend on the learnable matrix $\mathbf{K}_c$, and iteratively apply equation 10 until the desired accuracy is reached.

Next, we note that the projection shown in equation 9 uses Newton-like iterations which are very accurate, but can be rather expensive. We have found that for some problems, simple gradient descent iterations
$$\mathbf{z}_{k+1} = \mathbf{z}_{k+1}^* - \mathbf{K}_c^\top \mathbf{J}^\top \mathbf{c}(\mathbf{K}_c \mathbf{z}_{k+1}^*) \tag{13}$$
are sufficient for our purpose. For other applications we have also applied a few conjugate gradient steps to approximately solve the linear system equation 10.

The projection methods introduced in Section 2.2 and here both allow obeying the constraints to arbitrary precision, but they can be rather expensive due to the iterative projections. We next introduce a faster alternative.

### 2.4 STABILIZATION BY PENALTY

The last technique we explore is stabilization with respect to the additional information equation 3. Unlike the projection, stabilization does not aim to satisfy the constraints exactly but rather do it approximately throughout the network. To this end, note that a descent towards the constraint can be written as an approximation of an ODE of the form
$$\dot{\mathbf{z}} = -\mathbf{K}_c^\top \mathbf{J}(\mathbf{K}_c \mathbf{z})^\top \mathbf{H}\mathbf{c}(\mathbf{K}_c \mathbf{z}),$$
where $\mathbf{H}$ is any Symmetric Positive Definite matrix.

Therefore, one way to stabilize the system is to augment the original ODE with a term that "flows" towards the constraint. The resulting limit ODE becomes
$$\dot{\mathbf{z}} = \sigma(\mathbf{z}, \boldsymbol{\theta}) - \gamma\mathbf{K}_c^\top \mathbf{J}(\mathbf{K}_c \mathbf{z})^\top \mathbf{H}\mathbf{c}(\mathbf{K}_c \mathbf{z}). \tag{14}$$
Using again an explicit method to discretize the system we obtain a discrete analog that can be used to solve the problem.

In the following we use the RK4 discretization with $\mathbf{H} = \mathbf{I}$, which is often sufficient. But for some problems more advanced choices might be needed (Ascher, 1997). One common choice for $\mathbf{H}$ is $\mathbf{H} = (\mathbf{J}\mathbf{K}_c\mathbf{K}_c^\top \mathbf{J}^\top)^{-1}$.

## 3   MODEL PROBLEMS

In this section we discuss two model problems that are used to test the different architectures. The multi-body pendulum is a simple model problem that has a well studied numerical solution (Arnold, 2017). The ubiquitous molecular dynamics problem has received much attention in the literature (Allen et al., 2004; Kadupitiya et al., 2022; Schütt et al., 2018). Both problems are constrained mechanical systems with known equations of motion that can be solved by standard integration techniques. We use numerical simulations of these physical problems to generate datasets that allow us to determine and evaluate our constrained neural networks.

Our goal is to train a neural network to predict future states of mechanical systems given the present state.

### 3.1   THE MULTI-BODY PENDULUM

Our first mechanical system is a 2D multi-body pendulum, as shown in Figure 1. We chose this toy experiment since it has obvious constraints, as well as chaotic motion which makes its motion non-trivial to predict.

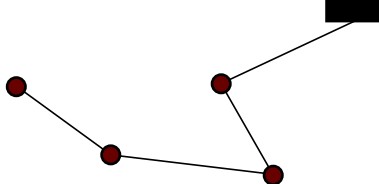

Figure 1: A multi-body pendulum system with four pendulums.

A multi-body pendulum can be parameterized in Cartesian coordinates by position $\mathbf{r} \in \mathbb{R}^{2 \times N}$ and velocity $\mathbf{v} \in \mathbb{R}^{2 \times N}$ matrices, where $N$ is the number of pendulums in the system. Since the distance between consecutive mass points is constant we have that at each time point along a trajectory

$$c_i = |\mathbf{r}_i - \mathbf{r}_{i-1}| - l_i, \quad i = 1, \ldots, N, \tag{15}$$

where $\mathbf{r}_0 = 0$, and $l_i$ is the length of the $i$th pendulum piece. The Jacobian of the constraint is easily calculated and it is a sparse matrix. In practice, only matrix-vector products with the Jacobian (and its transpose) are needed and this can be coded efficiently without storing the matrix.

### 3.2   WATER MOLECULES SIMULATION

For our second experiment we have created a microcanonical ensemble (NVE) simulation of 32 water molecules at a temperature of $300\,\mathrm{K}$, approximated with the Lennard-Jones force-field described in Praprotnik & Janežič (2005), using cp2k (Kühne et al., 2020). The physical simulation is done employing a step-size of $0.1\,\mathrm{fs}$ for $100,000$ steps. The simulation is done without periodic spatial boundaries in order to have a system that fundamentally evolves over time. Constraints are commonly added to MD simulations in order to freeze out high frequency vibrational movement of molecules, which enables the simulations to use larger time-steps in physical space (Allen et al., 2004).

Water molecules are known to be bound in a triangle configuration with $95.7\,\mathrm{pm}$ between the hydrogen and oxygen atoms and an angle of $104.5°$ between the hydrogen atoms as illustrated in Figure 2. Each water molecule is constrained separately and with identical constraints

$$c(\mathbf{r}_i, \mathbf{r}_j) = |\mathbf{r}_i - \mathbf{r}_j| - l_{ij}. \tag{16}$$

Note that while the pendulum constraint is exact, the water molecule constraint is not exact, and small deviations exist in the data due to wiggling/vibration of the water molecules around their bound state.

## 4   EXPERIMENTS

In Section 2, we discussed in consecutive subsections several ways for introducing auxiliary constraint information into a neural network, namely, **auxiliary loss**, **end constraints**, **smooth constraints**, and a **penalty** method. Here, we next evaluate these methods on the two model problems discussed in the previous section.

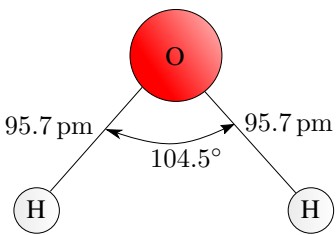

Figure 2: An illustration of the bound state of a water molecule.

## 4.1 MULTI-BODY PENDULUM

Our first experiment is the multi-body pendulum, described in Section 3.1. At the start of each experiment we simulate a pendulum for a sufficient number of timesteps to generate the amount of data samples needed for training, validation and testing the neural networks plus an additional $1000$ steps to ensure chaotic motion. The $i$-th data sample contains the positions and velocities of the system at steps $i$ and $i + k$. We randomly split the data into training, validation and testing datasets. When training the neural network on the $i$-th data sample, we use the input position and velocity as initial best guess, $\mathbf{x}_i = (\mathbf{r}_i, \mathbf{v}_i)$, and try to predict the future position $\mathbf{y}_i = \mathbf{r}_{i+k}$.

All multi-body simulations are done with a five-pendulum system, a step length of $0.01\,\mathrm{s}$, a pendulum mass of $1\,\mathrm{kg}$, a pendulum length of $1\,\mathrm{m}$, and a free-fall acceleration constant of $9.82\,\mathrm{ms}^{-2}$. We implement our neural networks using RK4 stepping between the layers. The neural network is trained with a batch-size of 10. For the constraints, we use a maximum of 100 steepest descent projections to obey the constraints with an early stopping criterion if the max constraint violation is ever below $1 \times 10^{-4}\,\mathrm{m}$. In this experiment we can control how difficult the problem is, by changing $k$. Figure 3 shows two snapshots of a five-pendulum configuration.

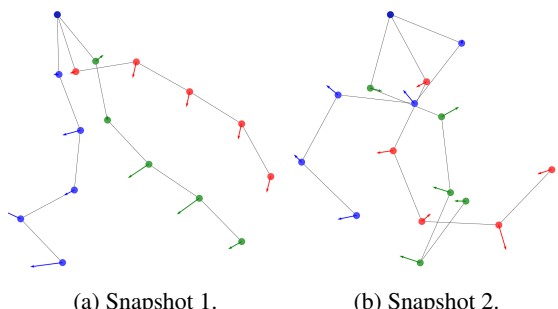

(a) Snapshot 1.  (b) Snapshot 2.

Figure 3: Snapshots of five-body pendulums. The green and blue pendulums follow the red pendulums by $k = 20$ and $k = 50$ steps, respectively. The arrows indicate velocity. Snapshot 1 shows the system before it fully enters chaotic motion, which takes about 1000 steps. Snapshot 2 shows the system at a later, more chaotic state.

We start by determining appropriate values for $\gamma$ and $\eta$ used in the penalty method and the auxiliary regularization method (For more details see Appendix B-C). We then compare the different ways of adding constraints to a neural network. Figure 4 shows the results of training a mimetic neural network without and with constraints (Eliasof et al., 2022).

Based on these results, we can see that end constraints and smooth constraints lead to unstable training, which we suspect is due to the fact that the neural network no longer predicts solutions even remotely close to obeying the constraints. Instead the solutions are projected onto the constraints, which works well for small constraint violations, but leads to instabilities for large constraint violations. In order to counteract this, we want the network to still predict reasonable pendulum positions before constraints are applied, and to that effect we have two different methods we can apply. One is to also add a penalty term to the end- and smooth-constraints, which naturally leads the network towards a solution that is close to obeying the constraints, as described in Section 2.4. The second method is to add an auxiliary regularization term to the loss function as described in Section 2.1 based on the constraint violation before any projections onto the constraints are applied in the neural network.

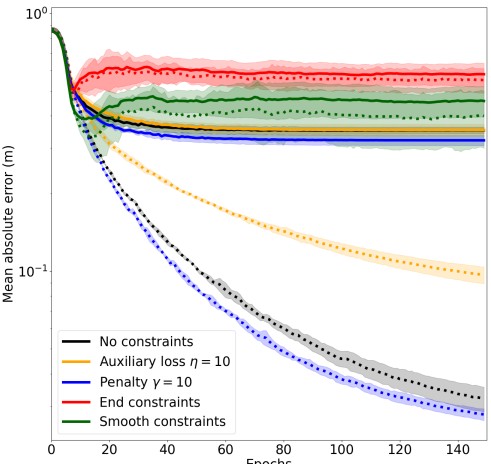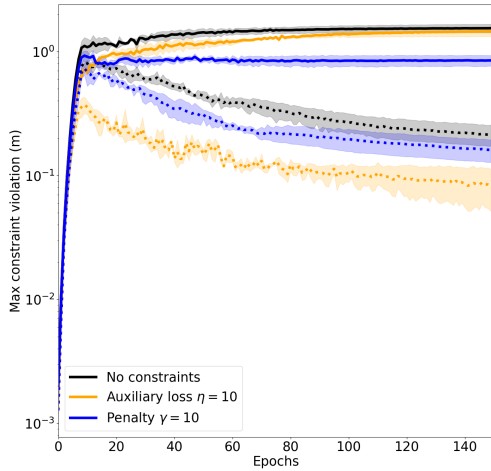

Figure 4: Comparison of the different ways of adding constraints to a neural network trained on 100 multi-body pendulum samples with $k = 20$. Left: Mean absolute error. Right: Maximum constraint violation. Constraint violations for end/smooth constraints are not shown since these are several orders of magnitudes lower and are determined by the number of projections/early stopping criteria. Each run has been repeated five times. The solid/dashed lines are the average based on validation/training data, while the shaded regions around each line designate error bounds of one standard deviation.

|  | Constraints | $n_t = 100$ | $n_t = 1000$ | $n_t = 10000$ |
|---|---|---|---|---|
|  | No constraints | 35.0 | 6.40 | 2.39 |
|  | Auxiliary loss $\eta = 10$ | 35.1 | 8.52 | 2.94 |
| $k = 20$ | Penalty $\gamma = 10$ (ours) | 32.6 | **5.72** | **2.32** |
|  | End constraints $\gamma, \eta = 10$ (ours) | 29.8 | 6.89 | 2.93 |
|  | Smooth constraints $\gamma, \eta = 10$ (ours) | **27.9** | 7.35 | 2.85 |
|  | No constraints | 67.6 | **10.7** | **3.98** |
|  | Auxiliary loss $\eta = 10$ | 78.0 | 31.8 | 27.4 |
| $k = 50$ | Penalty $\gamma = 10$ (ours) | **66.3** | 13.1 | 5.43 |
|  | End constraints $\gamma, \eta = 10$ (ours) | 68.8 | 40.9 | 48.8 |
|  | Smooth constraints $\gamma, \eta = 10$ (ours) | 73.2 | 40.3 | 52.7 |

Table 1: Mean absolute error (cm) over a test set of 1000 samples on the multi-body pendulum problem. $n_t$ is the number of training samples, while $k$ is the number of steps predicted ahead. Each experiment was repeated three times and the average of those runs are shown.

We test both methods separately with various strengths, and also a combination of the two, and find that the penalty method is insufficient to completely stabilize the training. Auxiliary loss on the other hand does stabilize the network if applied with sufficient strength. For both end constraints, and smooth constraints, we find that a combination of auxiliary loss $\eta = 10$ and penalty $\gamma = 10$ gives the best result. We still refer to these combined methods as end constraints or smooth constraints, but specify a value for $\gamma, \eta$. With the modified smooth and end constraints, we rerun the initial comparative experiment as shown in Figure 5.

We have investigated the effect of training sample size as well as the difficulty of the problem and show the results of this in table 1. Figure 6 shows a comparative example of a multi-body pendulum prediction based of neural networks with different constraining schemes.

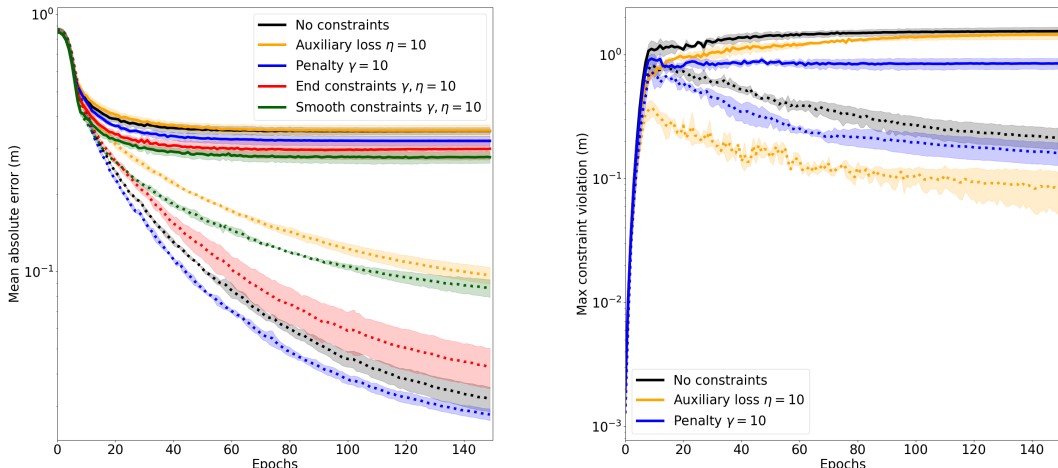

Figure 5: Comparison of the different ways of adding constraints to neural network trained on 100 multi-body pendulum samples with $k = 20$. Left: Mean absolute error. Right: Maximum constraint violation. Constraint violations for end/smooth constraints are not shown since these are several orders of magnitudes lower and determined by the number of projections/early stopping criteria. Each run has been repeated five times, the solid/dashed lines are the average based on validation/training data while the shaded regions around each line designate error bounds of one standard deviation.

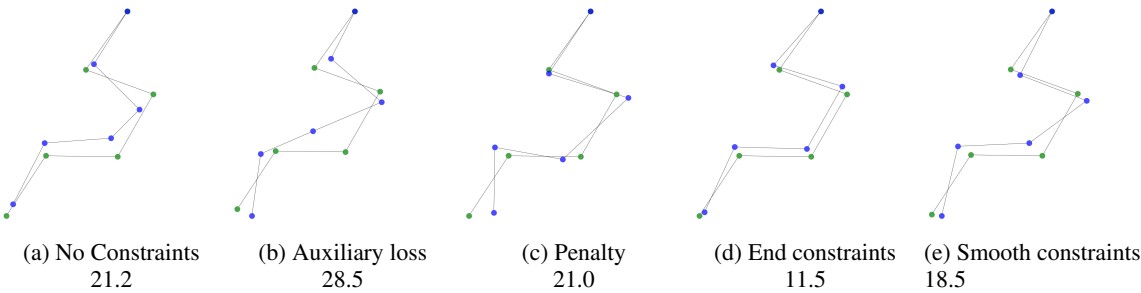

| (a) No Constraints | (b) Auxiliary loss | (c) Penalty | (d) End constraints | (e) Smooth constraints |
|---|---|---|---|---|
| 21.2 | 28.5 | 21.0 | 11.5 | 18.5 |

Figure 6: Comparative snapshots of the different trained networks during test evaluation. The networks are trained with $k = 20$ on 1000 training samples. The green pendulum is the desired output, while the blue pendulums are the predicted output. For auxiliary loss, end constraints and smooth constraints $\eta = 10$. For penalty, end constraints and smooth constraints $\gamma = 10$. The number given below each pendulum is the mean absolute error (cm) for that particular prediction.

## 4.2 MOLECULAR DYNAMICS SIMULATIONS - WATER MOLECULES

Our second experiment is the water molecule simulation described in Section 3.2. Similarly to the multi-body pendulum experiment, the $i$th data sample contains positions and velocities of the system at $i$ and $i + k$ steps. At the start of each experiment we randomly generate training/validation/testing datasets based on the $100,000$ physical time steps already simulated. When training the neural network on the $i$th data sample, we use the input position and velocity vectors as initial best guess, $\mathbf{x}_i = (\mathbf{r}_i, \mathbf{v}_i)$, and try to predict the future position $\mathbf{y}_i = \mathbf{r}_{i+k}$. Due to the nature of the simulation we change from a mimetic neural network to an SO3-equivariant neural network using the e3nn framework (Batzner et al., 2022). The neural network is trained with a batch-size of 10. For the smooth constraints, we use a maximum of 100 steepest descent projections and an early stopping of $5 \times 10^{-4}$ Å.

| | Constraints | $n_t = 100$ | $n_t = 1000$ | $n_t = 10000$ |
|---|---|---|---|---|
| | No constraints | 6.80 | 6.67 | 3.79 |
| | Auxiliary loss $\eta = 10$ | 6.82 | 6.70 | 3.28 |
| $k = 50$ | Penalty $\gamma = 10$ (ours) | 4.59 | 4.05 | **1.52** |
| | End constraints $\gamma, \eta = 10$ (ours) | 3.63 | 3.62 | 3.13 |
| | Smooth constraints $\gamma, \eta = 10$ (ours) | **3.55** | **3.50** | 3.18 |

Table 2: Mean absolute error (pm) over a test set of 1000 samples on the water molecule problem. $n_t$ is the number of training samples, while $k$ is the number of steps predicted ahead. Each experiment was repeated three times and the average of those runs are shown.

We use the modified versions of smooth-constraints and end-constraints with similar hyper parameters as in the previous experiment. Table 2 shows a comparison of the different constraint methods with sizes of training data.

## 5  DISCUSSION & CONCLUSION

In this work we have proposed different ways of incorporating constraint information in a neural network and successfully demonstrated their effectiveness on two separate modelling problems, namely, the multi-body pendulum, where the constraints are exact, as well as the more realistic water molecules simulation where the constraints are only approximately represented in the data.

Based on the results in Tables 1 and 2 we see that including constraints generally gives significant inference benefits. Projection constraints are often the best choice when the amount of training data is low, while the penalty method seems to be the best choice for large amounts of training data.

The explicit incorporation of constraints information into neural networks does not appear to be as essential a component to success as it is for classical MD simulations, likely due to the fact that this information is already inherently present in the data that the networks are learning from. Constraints do nonetheless bring significant improvements, especially in cases where there is limited training data. Furthermore, for certain fields, such as robotic movements or molecular dynamic propagation our work could be valuable, not just because the constraints might lead to better fitting on train/test data, but also because any constraint violation could lead to critical problems in such fields.

Our method uses RK4 stepping in $\tau$ combined with gradient descent projections, which was sufficient for the problems we have investigated thus far. But future studies using other stepping methods as well as Newton projections should be undertaken. Similarly, we set $\mathbf{H} = \mathbf{I}$ in the penalty term defined in Section 2.4, but know that more advanced expressions are commonly used in traditional DAEs. Performance-wise, while the constraint projection methods in their current form do add a significant performance booster, they could be optimized significantly. However, the stabilization method which only adds a simple Jacobian penalty term, can be used with very little in terms of performance degradation, and can lead to significantly improved results as seen in Tables 1 and 2.

## DATA AVAILABILITY

The code and data sets will be made available upon publication.

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

## A   NETWORK ARCHITECTURE

In this work we used two different neural networks, a 3D rotation-translation equivariant network and a non-equivariant mimetic network (Eliasof et al., 2022). Both networks are residual graph convolutional neural networks, with a learnable stepsize which starts out very small, which ensures that the initial network output is very similar to the best guess.

### A.1   EQUIVARIANT NETWORK

Our equivariant network is written using the e3nn software (Geiger et al., 2021), which allows for any geometric tensor to be accurately represented, and allows meaningful interactions between any such objects (Thomas et al., 2018). Our network is inspired by Batzner et al. (2022), and reaches comparable results to the ones shown in their work on MD17. The propagation block for our equivariant network is a simple equivariant convolutional filter, a non-linear activation, and a self-interacting tensorproduct. In this work we limit our network to scalars, pseudo-scalars, vectors, and pseudo-vectors. Our equiavariant network has $\approx 2.8M$ parameters and 8 layers.

### A.2   NON-EQUIVARIANT MIMETIC NETWORK

Our non-equivariant network is a mimetic network, with a propagational block which computes node averages and gradients as well as higher order products by moving node information through their edge connections. The information transfer between nodes and edges is done in a mimetic fashion (Lipnikov et al., 2014). Our mimetic network has $\approx 5.8M$ parameters in 8 layers.

## B   DETERMINING PENALTY STRENGTH

The penalty stabilization method mentioned in Section 2.4 contains the hyperparameter $\gamma$, which determines the strength of the penalty. In Figure 7, we set $\mathbf{H} = \mathbf{I}$ and test various values of $\gamma$ in order to find an appropriate value for it. Note that for this method the constraint violation does not simply diminish when $\gamma$ is increased.

## C   TRAINING WITH AUXILIARY REGULARIZATION

The stabilization method mentioned in Section 2.1 employs auxiliary regularization, which contains the hyper parameter $\eta$ that determines the strength of the regularization. With auxiliary regularization it is easy to minimize constraint violation, but it does so at the cost of overall learning as seen in Figure 8.

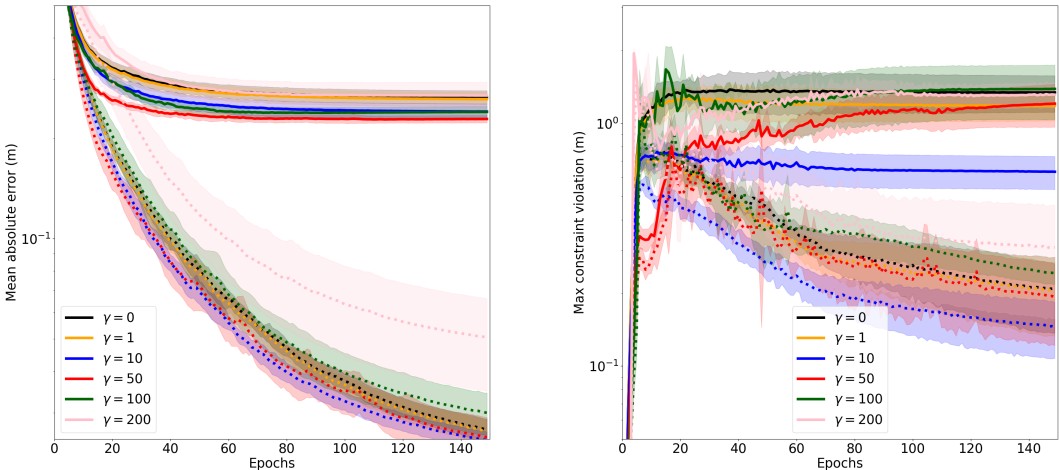

Figure 7: Comparison of neural networks using penalty stabilization with different strengths $\gamma$. All neural networks are trained on 100 samples with $k = 20$ on the multi-body pendulum system described in 4.1. Each run has been repeated five times, the solid/dashed lines are the average based on validation/training data while the shaded regions around each line designate error bounds of one standard deviation.

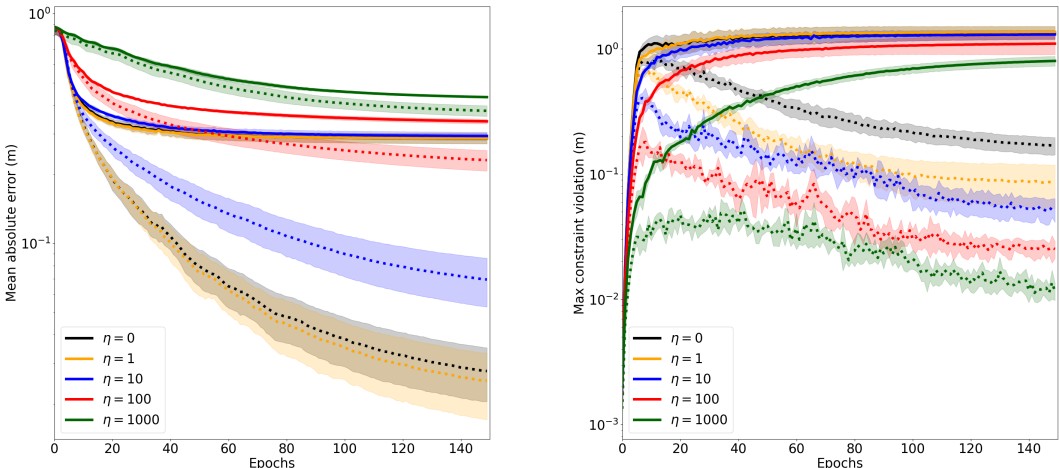

Figure 8: Comparison of neural networks using auxiliary regularization with different strengths $\eta$. All neural networks are trained on 100 samples with $k = 20$ on the multi-body pendulum system described in 4.1. Each run has been repeated five times, the solid/dashed lines are the average based on validation/training data while the shaded regions around each line designate error bounds of one standard deviation.

