# OpenReview forum: "Neural DAEs: Constrained neural networks"
_ICLR.cc/2023/Conference — Submitted to ICLR 2023_

### Official Review · Reviewer_vpjY · 2022-10-17

**Confidence:** 4
**Correctness:** 2
**Technical Novelty And Significance:** 2
**Empirical Novelty And Significance:** 2
**Recommendation:** 3

**Clarity, Quality, Novelty And Reproducibility:**

The motivation is clear, but it is unclear how the proposed method solves the problems. Novelty is limited. The experimental setting is unclear, so the reproducibility is limited.

**Strength And Weaknesses:**

Strength:

The motivation is clear.

Weaknesses:

While four methods are proposed, the difference between them is unclear. They are compared only by using experiments, and the motivation to propose them separately is unclear.

Many previous studies have tackled to satisfy constraints, but none of them is discussed or compared.

The definition of neural network architecture in (2) is awkward. $\sigma$ is not defined explicitly. If it is a neural network or some nonlinear function, (2) is equivalent to Neural ODE sandwiched by two projections by $K_0$ and $K_c$. If it is just an activation function, I cannot understand what (2) does mean. Anyway, it is never a residual network. The relationship with previous studies is unclear.

A detailed derivation of the first method is omitted and the reader is directed to the reference. In contrast, the detailed explanation about 4th-order Runge-Kutta method is provided. The organization is not reader-friendly.

According to Table 1, in many cases, the case with no constraints demonstrated better performances. So, the contribution of the proposal is unclear.

Details about experiments are missing, so it is hard to reproduce experiments. Both experiments involve only constraints that come from links between masses (or molecules). This constraint is so-called holonomic constraint. However, in the introduction, the authors mentioned other constraints such as the divergence of the magnetic field. Hence, the experiments are insufficient or the introduction is excessive advertising.

It appears to violate the format (namely, smaller margins used).

**Summary Of The Paper:**

This study proposes a neural network for differential algebraic equation (DAE). It is an ordinary differential equation (ODE) with algebraic constraints. The proposal is based on projection, and composed of four implementations.


**Summary Of The Review:**

The motivation is interesting, but the experiments do not adequately answer the motivation. The manuscript is not ready for publication.

---

### Official Review · Reviewer_Ypc6 · 2022-10-24

**Confidence:** 4
**Correctness:** 3
**Technical Novelty And Significance:** 2
**Empirical Novelty And Significance:** 2
**Recommendation:** 3

**Clarity, Quality, Novelty And Reproducibility:**

The idea of adding constraints to a given neural ODE in the paper is interesting and original. However, it lacks theoretical support and comparison with existing methods in the literature.

**Strength And Weaknesses:**

*Strengths*
- The motivation for the constrained neural network inherited from natural invariants of dynamical systems is clear and the experiments are interesting.
- The methods are derived in detail and easy to understand.

*Weaknesses*
- The paper lacks empirical theory. Effects of the proposed 4 methods are discussed based on experiments and there is no theorem/proposition analysing any properties of the methods.
- There is no comparison with any of the previous methods in the literature. Therefore, it is hard to evaluate the novelty of the results.


**Summary Of The Paper:**

The paper presents 4 different methods to explicitly adding auxiliary trajectory information to neural networks for dynamical systems rather than having the network implicitly learn it. Two experiments are conducted to verify that, depending on the amount of the data, these methods give significant inference benefit.

**Summary Of The Review:**

In many simulations of dynamical systems, there are natural invariants that we can describe by constraints. Inherited from this fact, the authors present 4 different methods to explicitly adding auxiliary trajectory information to neural networks for dynamical systems rather than having the network implicitly learn it. The methods are: adding a term to the loss function to maintain the constraints, projecting the final state or every immediate state to the constraint manifolds, stabilizing by penalty. Two experiments with the multi-body pendulum and water molecule are derived. Depending on the amount of the data, these methods give significant inference benefits.

However, the paper lacks theoretical justifications for the effects of the methods to the dynamical systems and training. The experiments are weak and the authors have not compared the current methods with previously known methods in the literature as mentioned in the Introduction. Therefore, I believe that the authors need to work further on the improvement of the papers to make it suitable for publication.

---

### Official Review · Reviewer_ku9X · 2022-10-26

**Confidence:** 3
**Correctness:** 3
**Technical Novelty And Significance:** 3
**Empirical Novelty And Significance:** 3
**Recommendation:** 5

**Clarity, Quality, Novelty And Reproducibility:**

Overall, the paper is well written and the idease and methods are well-presented. To the best of my knowledge, the proposed methods appear to be novel. The authors refer to earlier work throughout the paper and relation to the existing literature is made sufficiently clear (though I also believe that an explicit related work section clould have further improved this aspect). The authors also state that code will be made public upon acceptance, which clearly helps reproducibility.

**Strength And Weaknesses:**

## Strengths
- The setup is well-presented and well-motivated, and the derivation of the multiple approaches follows very clearly from the initial presentation (eqs. 2 and 3).
- The proposed methods seem conceptually simple and easy to implement, which increases their potential utility for practitioners that use neural ODEs.
(The "Clarity, Quality, Novelty and Reproducibility" section also lists further strengths)


## Weaknesses & Questions
My main concerns relate to the experimental evaluation:
- Experiments - combining auxiliary and penalty loss: The experimental findings indicate that for the "end constraints" and "smooth constraints" methods, additional losses are necessary to get good performance, and that a combination of auxiliary loss and penalty gives the best result. But using _only_ this combination is not experimentally evaluated, which would be, I believe, necessary to make any claims about the performance of the "end constraints" and "smooth constraints" methods (since the performance could also come already from just the combination of the auxiliary loss and penalty). This is I believe a necessary evaluation to add, to be able to claim that "the constraint projection methods [...] add a significant performance booster".
- Experiments - interpreting the results: I find it hard to evaluate the results presented in table 1. In table 2, I can observe that the error is roughly halved by using the penalty and/or constraints, compared to the two baselines, and I also observe that the scale of error does not change too much between the three setups. But in table 1, errors range from 66 to 2, which makes me question the relevance of the low-data experiments: If all methods perform "badly" for $n_t=100$ and $k=50$, as in they all perform close to random, can we really compare their performance and state that some perform better than others? Maybe this could be clarified by stating relative errors (to make the numbers shown more interpretable) and by adding error ranges (and according to Figures 4 and 5 experiments were performed for multiple seeds anyways).
  Clarifying this point is particularly releavant since the paper claims that "results in tables 1 and 2 [show] that including constraints generally gives significant inference benefits".


### Minor things
- Section 2, "Next, the residual network uses m layers with learnable parameter \theta": This sentence is not clear from the context as "m" does not appear anywhere. Please clarify.
- Section 2.3 appeared to be a bit misleading. It performs a specific derivation with Lagrange multipliers which ends up at equation 9; only to then consider projections in general (eq. 11c & 12), which are then done in multiple ways. It might be helpful for clarity to mention projections earlier (which are already established in section 2.2) and instead discuss the choice of projection method more explicitly (since the method of choice seem to be gradient descent steps; this further motivates that deriving equation 9 might not be necessary since eq. 13 is used instead); and possibly compare this choice experimentally.
- On "Neural DAEs": I am not quite convinced that this is an appropriate name for the proposed methodology. "Neural ODEs" can be understood as ODEs $y' = f_\theta(y)$, where $f_\theta$ is a neural network. A natural way to think of "neural DAEs" would then be to consider DAEs, e.g. of the form $y' = f_\theta(y, z)$, $0 = g_\theta(y, z)$, and have both $f_\theta$ and $g_\theta$ be neural networks; that is, the constaint would also be learned.


**Summary Of The Paper:**

The paper investigates methods for including known constaints into neural ODEs. It presents new approaches based on projection and ODE augmentation, which are compared and evaluated on two k-step-ahead prediction tasks. The results show some benefits of including such constraints, in particular in the low-data regime.


**Summary Of The Review:**

The paper considers the interesting and relevant setup of dynamical systems that satisfy known constraints, and aims at improving learned neural networks by including these constraints into the training regime, by using penalties and projections.

Unfortunately, at the current state the paper does not fulfill its claim that the experimental results show "that including constraints generally gives significant inference benefits". This is due to (i) difficulties in understanding the "significance" of the results, in particular of table 1, and (ii) the lacking evaluation of the auxiliary+penalty combination, without which claims about the relevance of the projection methods is not possible.

I therefore lean towards rejection.

---

### Decision · Program_Chairs · 2023-01-20

**Decision:**

Reject

**Justification For Why Not Higher Score:**

See above.

**Justification For Why Not Lower Score:**

N/A

**Metareview: Summary, Strengths And Weaknesses:**

All three reviewers had concerns, including major concerns, with this paper, regarding clarity and novelty. The authors did not respond to the reviews. A clear reject